# Total plaque score helps to determine follow-up strategy for carotid artery stenosis progression in head and neck cancer patients after radiation therapy

Chi-Hung Liu[1‡], Joseph Tung-Chieh Chang[2,3,4‡], Tsong-Hai Lee[1], Pi-Yueh Chang[5,6], Chien-Hung Chang[1], Hsiu-Chuan Wu[1], Ting-Yu Chang[1], Kuo-Lun Huang[1], Chien-Yu Lin[2,3,4,7], Kang-Hsing Fan[2,3,4], Chan-Lin Chu[1], Yeu-Jhy Chang[1,8]*

1 Stroke Center and Department of Neurology, Chang Gung Memorial Hospital, Linkou Medical Center and College of Medicine, Chang Gung University, Taoyuan, Taiwan, 2 Department of Radiation Oncology, Proton and Radiation Therapy Center, Chang Gung Medical Foundation, Linkou Chang Gung Memorial Hospital, Taoyuan, Taiwan, 3 Taipei Chang Gung Head & Neck Oncology Group, Chang Gung Memorial Hospital Linkou Medical Center, Taoyuan, Taiwan, 4 Department of Medicine, College of Medicine, Chang Gung University, Taoyuan, Taiwan, 5 Department of Laboratory Medicine, Chang Gung Memorial Hospital, Linkou Medical Center, Taoyuan, Taiwan, 6 Department of Medical Biotechnology and Laboratory Science, Chang Gung University, Taoyuan, Taiwan, 7 Particle Physics and Beam Delivery Core Laboratory of Institute for Radiological Research, Chang Gung University/Chang Gung Memorial Hospital, Linkou, Taoyuan, Taiwan, 8 Chang Gung Medical Education Research Centre, Taoyuan, Taiwan

‡ These authors contributed equally and are co-first authors.
* yjc0601@cgmh.org.tw

**Data Availability Statement:** Data cannot be completely shared publicly because of the regulations from our Ethics Committee. Data

## Abstract

### Background

To identify predictors of carotid artery stenosis (CAS) progression in head and neck cancer (HNC) patients after radiation therapy (RT).

### Methods

We included 217 stroke-naïve HNC patients with mild carotid artery stenosis after RT in our hospital. These patients underwent annual carotid duplex ultrasound (CDU) studies to monitor CAS progression. CAS progression was defined as the presence of ≥50% stenosis of the internal/common carotid artery on follow-up CDU. We recorded total plaque score (TPS) and determined the cut-off TPS to predict CAS progression. We categorized patients into high (HP) and low plaque (LP) score groups based on their TPS at enrolment. We analyzed the cumulative events of CAS progression in the two groups.

### Results

The TPS of the CDU study at enrolment was a significant predictor for CAS progression (adjusted odds ratio [aOR] = 1.69, p = 0.002). The cut-off TPS was 7 (area under the curve: 0.800), and a TPS ≥ 7 strongly predicted upcoming CAS progression (aOR = 41.106, p = 0.002). The HP group had a higher risk of CAS progression during follow-up (adjusted

inquiries can be sent to the Chang Gung Medical Foundation Biobank (Email: tissuebankcgmh@gmail.com). The relevant data which could be provided are within the manuscript and its Supporting information files.

**Funding:** Funding was provided by Chang Gung Memorial Hospital (grant number: CMRPG381503, CMRPG3C0763, CMRPG3G0261, BMRPF99) and the Ministry of Science and Technology (grant numbers: 106-2511-S-182A-002 -MY2, 108-2314-B-182A-050-MY3, NMRPG3G6411-2, and NMRPG3J6131-3).

**Competing interests:** The authors have declared that no competing interests exist.

hazard ratio = 6.15; 95% confident interval: 2.29–16.53) in multivariable Cox analysis, and also a higher trend of upcoming ischemic stroke (HP vs. LP: 8.3% vs. 2.2%, p = 0.09).

## Conclusions

HNC patients with a TPS ≥ 7 in any CDU study after RT are susceptible to CAS progression and should receive close monitoring within the following 2 years.

## Introduction

Head and neck cancer (HNC), and particularly nasopharyngeal cancer, has a unique geographic distribution, and it primarily occurs among Asian populations [1]. In these patients, radiation therapy (RT) or combined chemotherapy and RT is the current treatment standard [1]. With a significant increase in survival rates from HNC over the past two decades [2], addressing RT-related complications has become a major challenge following HNC remission.

The long-term consequences of radiation injury [3] include radiation-induced vasculopathy with accelerated atherosclerosis and an increased risk of carotid artery stenosis (CAS) [4, 5]. Compared to atherosclerosis-induced CAS, RT-induced CAS spreads extensively [6], progresses more rapidly [7], and typically affects the common carotid artery (CCA) [8]. Moreover, the risk of future ischemic stroke (IS) is also increased in these patients [9, 10]. The risk of in-stent restenosis after carotid artery stenting is also higher in patients with HNC than in those with atherosclerosis [11]. In contrast to patients with asymptomatic atherosclerosis, patients with HNC require more frequent monitoring for the appearance of radiation vasculopathy [12]. Therefore, effective vascular screening and monitoring strategies for HNC patients after RT are urgently needed [3, 4, 12]. In addition, such strategies should aid in identifying the patients who have a high risk of CAS progression.

Patients with moderate to severe CAS or prior IS are usually under close surveillance regardless of whether or not they have a history of RT. However, whether HNC patients with mild CAS should be monitored as closely as those with moderate to severe CAS remains uncertain. In the present study, we followed HNC patients after RT at our hospital using carotid duplex ultrasound (CDU). We aimed to identify practical and easy-to-use clinical predictors for CAS progression in these HNC patients with mild CAS. We hoped that such predictors could help to determine the patients who should receive closer monitoring in the following years.

## Materials and methods

### Patient and data recruitment

Between January 1, 2013 and December 31, 2014, we prospectively enrolled HNC patients who had completed RT at Linkou Chang Gung Memorial Hospital. In this study, we primarily focused on stroke-naïve patients with mild CAS in CDU studies. We aimed to identify patients who may be susceptible to CAS progression in the following years. Therefore, patients with ≥50% CAS at enrolment and those with prior IS were excluded. In addition, although the patients were asked to undergo annual CDU follow-up studies, those without any follow-up CDU data after 1 year were also excluded (Fig 1). The study was approved by the Ethics Institutional Review Board of Chang Gung Memorial Hospital (IRB No. 100-4153B). All of the included patients signed written informed consent forms.

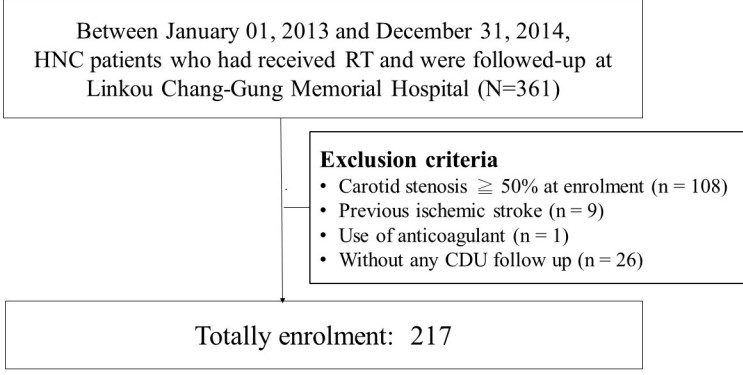

**Fig 1. Patient enrolment.** CDU, carotid duplex ultrasound; RT, radiation therapy; HNC, head and neck cancer.

Data on demographic and common stroke risk factors, including dyslipidemia, hypertension, diabetes mellitus, heart disease, and cigarette smoking, were obtained from all of the recruited patients. Laboratory data including baseline glycated hemoglobin, low-density lipoprotein cholesterol, and serum creatinine levels, as well as the use of antiplatelets or statins were also recorded. In addition, the type and stage of HNC, accumulated total dose of RT, and the time interval from the last RT (date of the last RT fraction) to study enrolment (date of the CDU study at enrolment) were also ascertained.

## Methods of RT

In the present study, a minimum of 5 mm around the clinical target volume was required in all directions to define each respective planning target volume, except for situations in which the gross target volume was adjacent to the brain stem. The treatment dose was 70 Gy/33 fx, which was delivered once daily, with 5 fractions per week, over 6 weeks and 3 days. All targets were treated simultaneously. Total treatment times more than 5 days longer than scheduled were considered to be major violations [13].

## Carotid duplex ultrasound studies

Philips HDI 5000 (Wesley Hills, NY, USA) or Acuson Sequoia (Siemens, Munich, Germany) 5 to 10 MHz real-time B-mode imaging systems and a 3.0 MHz pulsed-wave color Doppler spectrum analyzer were used in this study. Stenotic features in the sagittal (anterior–posterior, posterior–anterior, and lateral) and transverse views of the extracranial carotid arteries were analyzed during the B-mode examination. The degree of CAS was examined in the CDU study according to standard ultrasound criteria. The percentage of maximum stenosis in longitudinal views was determined using computer-assisted measurements of the 1-residual lumen diameter/vessel diameter × 100. The angle of insonation was set at 60 degrees during the flow velocity examination. Peak systolic velocities ≥120 cm/s were used to identify ≥50% CAS [14]. The investigators who performed and read (CLC et al.) the baseline and follow-up CDU were blinded to the patients' clinical conditions. Agreement was achieved in all CDU results between the investigators. Our CDU laboratory has an overall diagnostic accuracy rate of >90% for carotid stenosis [15, 16].

Each patient underwent the first CDU study at enrolment. We recorded the degree of CAS in each examined artery. Since total plaque score (TPS) is a well-known predictor of CAS after

RT [6], we also assessed the presence and severity of plaques in each CDU study [6]. We measured five segments including the proximal CCA, distal CCA, carotid bifurcation, internal carotid artery (ICA), and external carotid artery on each side. In total, we assessed 10 segments bilaterally. For each segment, we graded the plaques as follows: Grade 0, normal or no plaques; Grade 1, all plaques occupying <30% of the vessel diameter; Grade 2, at least one plaque occupying 30% to 49% of the vessel diameter; Grade 3, at least one plaque occupying 50% to 69% of the vessel diameter; Grade 4, 70% to 99% of the vessel diameter; and Grade 5, total occlusion of the vessel. Similar to a previous study, we defined the TPS for each patient as the sum of the plaque scores obtained from the five arterial segments in both carotid arteries [6]. We hoped to define a cut-off TPS to select HNC patients with mild CAS who may be vulnerable to CAS progression in the following years.

### Follow-up and outcomes

The main outcome of interest in this study was the presence of CAS progression. Therefore, serial CDU studies were performed annually to monitor CAS progression in the enrolled patients. We defined CAS progression as the presence of >50% stenosis on B-mode with a compatible hemodynamic pattern in any ICA or CCA on a follow-up CDU study.

### Statistical analysis

All data were retrospectively analyzed using SPSS version 22.0 (SPSS, Chicago, IL, USA). We wanted to identify factors that could predict CAS progression during the follow-up period. Cross-sectional analysis was first applied in the patients who completed their CDU follow-up within 2 years after enrolment. We used a multivariate logistic regression model to adjust for the confounding effects of the parameters that could predict CAS progression (age, smoking, history of hypertension, use of antiplatelets, use of statins, creatinine, glycated hemoglobin, low-density lipoprotein level, tumor stage, presence of lymph node invasion, type of HNC, time interval from the last RT, dose of RT, and TPS in model 1). We then used receiver operating characteristic analysis with a nonparametric model to determine the predictive accuracy of TPS following CAS progression. We used area under the curve analysis to assess the predictive accuracy. We used the Youden index as a criterion for deciding the optimal cut-off TPS in the CDU study at enrolment, and then repeated multivariate logistic regression analysis to examine the adjusted odds ratios (aORs) when the TPS was higher than the cut-off value in a CDU study (model 2).

The patients were further categorized into high plaque (HP) and low plaque (LP) score groups if their TPS in the CDU study at enrollment was higher or lower than the cut-off value, respectively. We used the independent two-sample *t* test to examine differences in continuous data between the HP and LP groups. In addition, categorical variables were compared using the chi-square test or Fisher's exact test. Survival analysis was then performed to determine whether TPS could predict CAS progression 2 years after enrolment. Cumulative events of time to CAS progression for the two groups were analyzed using a multivariable Cox model after adjusting for age, gender, smoking, hypertension, antiplatelets, creatinine, and serum uric acid (UA). Statistical significance was set at $p < 0.05$.

### Results

Between 2013 and 2014, we prospectively screened 361 Han Chinese patients with HNC who received RT at our hospital. Of them, 108 patients who had ≥50% CAS on the CDU study at enrolment were excluded. Furthermore, nine patients who had previous IS, one patient who

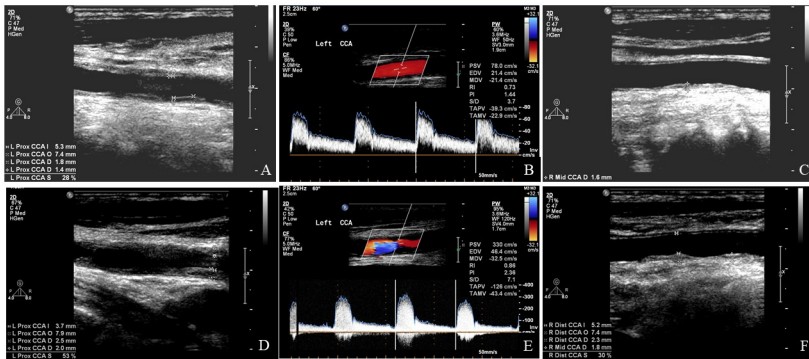

**Fig 2. An illustrated case with carotid artery stenosis progression.** This figure shows typical carotid duplex ultrasound findings in a 69-year-old male patient after radiation therapy. Prominent segmental changes were noted in bilateral common carotid arteries at the time of enrolment (A–C). Follow-up carotid duplex ultrasonography of the same patient 2 years later (D–F) demonstrated significant progression of stenosis (>50%) in the left common carotid artery (D) with hemodynamic changes (E).

received anticoagulant therapy, and 26 patients who did not receive any follow-up CDU studies were also excluded. Finally, we enrolled 217 eligible patients for further analysis (Fig 1).

Of these 217 patients, 209 (96.3%) received CDU follow-up for more than 18 months after enrolment. We first recorded the presence and evaluated the predictors of CAS progression (Fig 2). We adjusted for possible clinical confounding parameters in a multivariate logistic regression model (Table 1, model 1). Among these parameters, TPS in the CDU study at enrolment (aOR = 1.69; 95% confidence interval (CI): 1.21–2.37; p = 0.002) remained the most significant predictor for the presence of CAS progression. Other significant predictors included the time interval from the last RT to the CDU study at enrolment (aOR = 1.58; 95% CI: 1.05–2.36; p = 0.027) and age at enrolment (aOR = 0.86; 95% CI: 0.74–0.99; p = 0.041). The predictive accuracy of the TPS in the CDU study at enrolment was 0.800. When the cut-off TPS was

**Table 1. Multivariate logistic regression analysis of the predictors of carotid stenosis progression within the first 2 years.**

| | $\beta$ | aOR | $p$ |
|---|---|---|---|
| *Model 1*[*] | | | |
| Total plaque score at the first CDU | 0.53 | 1.69 | 0.002[‡] |
| Time interval from the last RT | 0.46 | 1.58 | 0.027[‡] |
| Age | -0.15 | 0.86 | 0.041[‡] |
| Triglyceride level | 0.01 | 1.01 | 0.065 |
| *Model 2*[†] | | | |
| Total plaque score $\geq$ 7 in the first CDU | 3.72 | 41.11 | 0.002[‡] |
| Time interval from the last RT | 0.48 | 1.61 | 0.033[‡] |
| Triglyceride level | 0.01 | 1.01 | 0.034[‡] |

CDU, carotid duplex ultrasound; RT, radiation therapy; HNC, head and neck cancer; aOR, adjusted odds ratio.

[*]*Model 1*: Age, smoking, history of hypertension, use of antiplatelets, use of statins, creatinine, glycated hemoglobin, low-density lipoprotein level, tumor stage, presence of lymph node invasion, type of HNC, time interval from the last RT, dose of RT, and total plaque score at first visit were adjusted in this model.

[†]*Model 2*: A total plaque score $\geq$ 7 in the first CDU and all parameters used in model 1 were adjusted in this model.

[‡]$p < 0.05$.

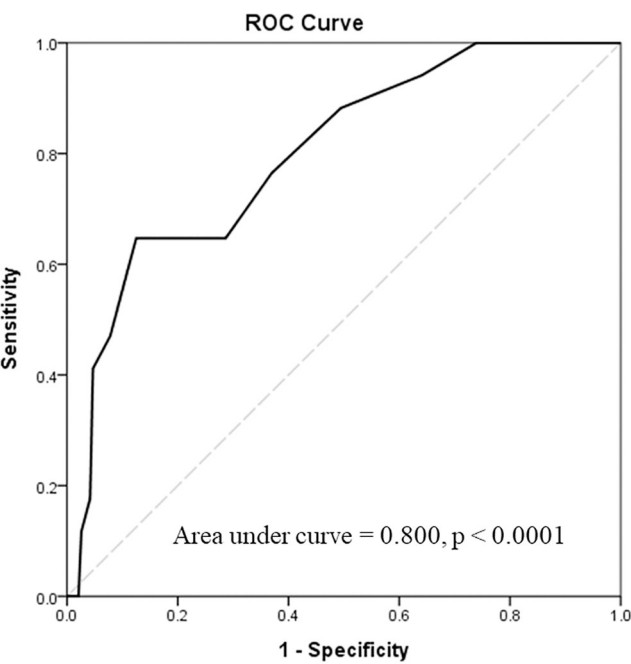

**Fig 3. Receiver operating characteristic analysis of the predictive accuracy of total plaque score on the presence of carotid stenosis progression.** The area under the curve of the total plaque score was 0.800 ($P < 0.001$), suggesting reasonable accuracy in predicting carotid stenosis progression within the following 2 years.

set at 7, the sensitivity and specificity to predict CAS progression were 0.647 and 0.875, respectively (Fig 3; p < 0.0001). After adjusting for other confounding factors in the multivariate logistic regression model (Table 1, model 2), TPS ≥ 7 in the CDU study at enrolment became a powerful clinical predictor (aOR = 41.106; 95% CI: 4.07–415.34; p = 0.002) for CAS progression.

Of the 217 patients, 36 (17%) had a TPS ≥ 7 in the CDU study at enrolment and were categorized into the HP group, and the other 181 (83%) patients in whom the TPS was < 7 were categorized into the LP group. The baseline characteristics of the two groups are shown in Table 2. Compared with the LP group, the HP group were older (HP vs. LP: 59.13±9.59 vs. 55.19±9.76 years, p = 0.03), male predominant (86.1% vs. 66.9%, p = 0.02), and were associated with higher frequencies of hypertension (52.8% vs. 30.4%, p = 0.01) and smoking (72.2% vs. 52.5%, p = 0.03). Moreover, the HP group had worse renal function (creatinine; HP vs. LP: 0.99±0.43 vs. 0.86±0.26 mg/dL, p = 0.02) and higher frequency of antiplatelet use (97.2% vs. 58.0%, p < 0.01). However, the mean total RT dose, time interval from the last RT, and the frequencies of nasopharyngeal carcinoma and advanced cancer stages (T3 or T4) were similar between the two groups (Table 2).

The follow-up durations were similar between the two groups in this study (HP vs. LP: 138.72±46.01 vs. 147.35±45.60 weeks, p = 0.30). During the follow-up period, the HP group had a higher frequency of CAS progression (HP vs. LP: 43.3% vs. 5.8%, p < 0.001). The multivariate-adjusted survival curves exhibited a higher risk of CAS progression in the HP group than in the LP group (adjusted hazard ratio = 6.15; 95% CI: 2.29–16.53, p < 0.001; Fig 4). In addition, the HP group had a higher trend of future IS (HP vs. LP: 8.3% vs. 2.2%, p = 0.09). However, the frequencies of death (HP vs. LP: 5.6% vs. 2.2%, p = 0.26) and tumor recurrence (HP vs. LP: 8.3% vs. 8.9%, p = 1.00) were not different between the two groups.

**Table 2. Baseline characteristics between the high (HP) and low plaque (LP) score groups.**

|  | HP group | LP group | *P* |
|---|---|---|---|
|  | (N = 36) | (N = 181) |  |
| Demographics |  |  |  |
| Age (years) | 59.13±9.59 | 55.19 ± 9.76 | 0.03 |
| BMI (kg/m$^2$) | 25.13±3.41 | 24.64 ± 4.61 | 0.55 |
| Gender (male, %) | 31 (86.1%) | 121 (66.9%) | 0.02 |
| Hypertension (%) | 19 (52.8%) | 55 (30.4%) | 0.01 |
| Diabetes mellitus (%) | 6 (16.7%) | 30 (16.6%) | 0.99 |
| Smoking (%) | 26 (72.2%) | 95 (52.5%) | 0.03 |
| NPC (%) | 19 (52.8%) | 115 (63.5%) | 0.23 |
| T3 or T4 stage | 15 (44.1%) | 66 (37.9%) | 0.50 |
| RT dose (cGy) | 6956.71±424.30 | 6952.86 ±387.06 | 0.96 |
| RT interval (years) | 8.81±4.66 | 9.56 ±3.67 | 0.37 |
| Laboratory data |  |  |  |
| HbA1C (%) | 5.88 ±0.66 | 5.86 ±0.58 | 0.90 |
| Cr (mg/dL) | 0.99 ±0.43 | 0.86±0.26 | 0.02 |
| LDL (mg/dL) | 123.20±30.70 | 121.50±41.85 | 0.78 |
| Triglyceride (mg/dL) | 144.66±89.90 | 127.12±83.57 | 0.26 |
| Serum UA (mg/dL) | 6.54±1.50 | 5.84±1.44 | 0.01 |
| Medications |  |  |  |
| Anti-platelets (%) | 35 (97.2%) | 105 (58.0%) | <0.01 |
| Statins (%) | 5 (13.9%) | 24 (13.3%) | 0.92 |
| Follow-up duration (weeks) | 138.72±46.01 | 147.35±45.60 | 0.30 |

BMI, body mass index; NPC, nasopharyngeal carcinoma; RT, radiation therapy; HbA1C, glycated hemoglobin; Cr, creatinine; LDL, low-density lipoprotein; UA, uric acid.

*High plaque score group included patients with a total plaque score ≥7 in the first carotid duplex ultrasound study at enrolment; low plaque score group included patients with a total plaque score <7 in the first carotid duplex ultrasound study at enrolment.

## Discussion

Progressive CAS is a major long-term complication in patients with HNC after RT, and may lead to a higher incidence of future IS [17]. Previous studies have demonstrated some independent predictors for CAS progression [6, 18], however it remains undetermined whether all patients should receive the same follow-up strategy. In patients with mild CAS, a convenient and useful clinical parameter that can help to identify patients who should receive intensive monitoring is particularly important. Our data showed that patients with a higher TPS (≥ 7) on the CDU study at enrolment were vulnerable to CAS progression in the following 2 years. This finding provides practical guidance for a follow-up strategy in HNC patients with mild ICA or CCA stenosis. In patients with a higher TPS, a closer follow-up CDU plan is necessary even when the time interval from the last RT is short. In contrast, in patients with a lower TPS, the next follow-up CDU can be delayed even when the time interval from the last RT is long.

Radiation vasculopathy is well known to have a distinct pathogenesis compared with atherosclerosis. Inflammatory processes, vasogenic edema on the endothelium, intraplaque hemorrhage, endothelial proliferation, adventitia fibrosis, and vasa vasorum occlusion may play crucial roles in radiation-induced CAS [19]. Therefore, the mechanism of CAS progression in radiation vasculopathy appears to be complex [20]. Although atherosclerosis per se can worsen the progression of CAS [19], continued vascular remodeling may also be another causative

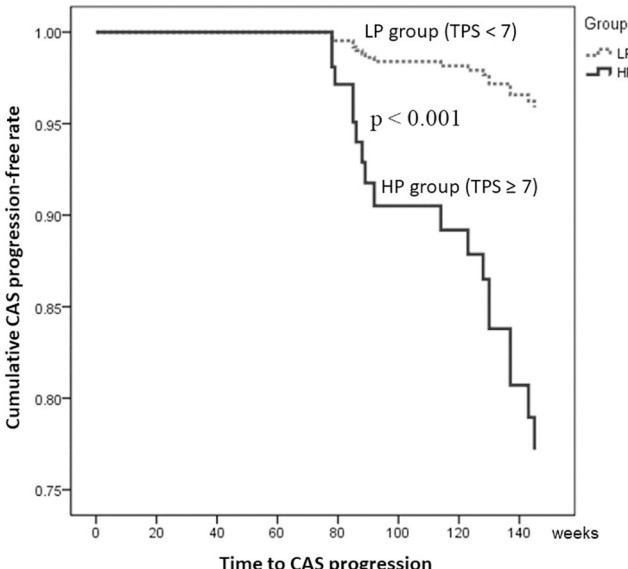

**Fig 4. Multivariate-adjusted survival curves estimating the time to CAS progression.** The incidence rate of CAS progression was significantly higher in the HP group than in the LP group. CAS, carotid artery stenosis; HP, high plaque score; LP, low plaque score; TPS, total plaque score.

factor leading to CAS progression [19]. The prevention of IS should be the final goal in HNC patients with radiation vasculopathy. In the present study, the HP group had a higher trend of upcoming IS and a higher frequency of antiplatelet use. Aspirin is known to prevent athero-sclerosis through anti-inflammatory and inhibitory effects on platelet adhesion and aggrega-tion [21]. However, radiation-induced atherosclerosis could not be successfully prevented by aspirin, clopidogrel, or statins in previous animal studies [22, 23]. A nationwide cohort study also precluded the protective effect of oral antithrombotic therapy on primary stroke preven-tion in HNC patients after RT [24]. Similar to these studies, our data were insufficient to show whether the use of antiplatelets was helpful to prevent IS. Statins have been reported to have anti-atherosclerotic effects [25], and dyslipidemia has been associated with increased intima thickness in HNC patients after RT [26]. However, the use of statins was not a protective factor for CAS progression in our multivariate model. The initial triglyceride level had a minimal predictive value for CAS progression in our data. However, we did not strictly control the use of fibrates, and therefore our results were insufficient to demonstrate the clinical significance of triglyceride level. Future studies are warranted to develop more effective preventive treat-ment in these HNC patients after RT.

In this study, the HP group had a higher serum UA level than the LP group. Serum UA is the final oxidation product of purine catabolism. Previous studies have shown that elevated serum UA levels may be associated with arterial stenosis and endothelial dysfunction [27, 28]. In addition, elevated serum UA was shown to contribute to the progression of atherosclerosis and arterial occlusion in a rat model [29]. However, it remains unclear whether the increase in serum UA levels is a compensatory mechanism to counteract oxidative stress or a marker reflecting the reactive oxygen species generated during the catabolism of purine [27]. More-over, serum UA may be released by apoptotic cells, suggesting a process of cellular injury and rapid cell turnover [30]. Serum UA has also been reported to be a marker representing higher tumor burden and clinical staging, tumor progression, and mortality in patients with HNC

[30]. In our data, either the T3 or T4 stage of HNC (p = 0.10) or the presence of lymph node invasion (p = 0.23) could predict CAS progression in the multivariate logistic regression model. There were no significant differences in advanced cancer stage, lymph node involvement, tumor recurrence or death between the HP and LP groups in this study. Therefore, the higher UA level in the HP group might not be related to higher tumor burden. Our results are insufficient to show whether the higher UA level in the HP group was associated with more severe endothelial dysfunction and endothelial activation. Further studies focusing on endothelial dysfunction and proliferation in radiation vasculopathy are warranted.

HNC is a characteristic disease in Chinese populations [31], and thus it was easier to recruit more patients and enhance the generalizability of our study results. However, there are still several limitations to this study. First, CDU studies may not be a precise method to evaluate the degree of stenosis and plaque features. Confounding factors caused by the technique may have influenced the study results, and thus limited the interpretation of CAS progression. In addition, advanced cerebrovascular images including computerized tomography angiography, magnetic resonance angiography, or conventional angiography were not routinely arranged as a confirmatory study when CAS progression was detected by CDU. This may also have limited the interpretation of the study results. Second, baseline vascular status may have confounded the conclusions drawn from our results. A higher TPS or CAS before RT may have further worsened the clinical progression and may have led to bias in this study. However, in real-world practice, primary physicians may not know the vascular status of the carotid arteries prior to RT. Further cohort studies with more strict follow-up protocols and confirmatory vascular imaging studies are still needed to validate our suggested follow-up strategy. Third, genetic factors among the study population were not assessed, and thus we could not demonstrate whether the HNC patients with damaged DNA repair were susceptible to CAS progression after RT [32]. Fourth, the single-center nature of this study could have led to patient selection bias. Fifth, the generalizability of our conclusions to other ethnicities remains uncertain. Lastly, a longer follow-up duration may provide more conclusive answers. Completely shifting practice paradigms based on what our preliminary results should be done only with immense circumspection.

In the future, we may need a cross reference with advanced structural cross-sectional angiographic imaging, a longitudinal data to assess true progression over a longer period of time, and more comprehensive baseline variables in these patients for further clarification of our conclusions.

## Conclusion

Our results may help guide healthcare professionals to tailor the follow-up strategies in HNC patients with mild CAS after RT. TPS is a practical and powerful parameter to predict CAS progression, and patients with a TPS $\geq 7$ on any CDU study should receive close monitoring in the following 2 years.

## Supporting information

**S1 Data.**
(XLSX)

## Acknowledgments

The authors thank Chang Gung Memorial Hospital, the Ministry of Science and Technology and Ms. Elaine Shinwei Huang for their administrative work.

## Author Contributions

**Conceptualization:** Chi-Hung Liu, Joseph Tung-Chieh Chang, Tsong-Hai Lee, Yeu-Jhy Chang.

**Data curation:** Chi-Hung Liu, Joseph Tung-Chieh Chang, Tsong-Hai Lee, Pi-Yueh Chang, Chien-Hung Chang, Ting-Yu Chang, Kuo-Lun Huang, Chien-Yu Lin, Kang-Hsing Fan, Chan-Lin Chu, Yeu-Jhy Chang.

**Formal analysis:** Chi-Hung Liu, Joseph Tung-Chieh Chang, Yeu-Jhy Chang.

**Funding acquisition:** Chi-Hung Liu, Yeu-Jhy Chang.

**Investigation:** Chi-Hung Liu, Joseph Tung-Chieh Chang, Hsiu-Chuan Wu, Kuo-Lun Huang, Yeu-Jhy Chang.

**Methodology:** Chi-Hung Liu, Joseph Tung-Chieh Chang, Tsong-Hai Lee, Pi-Yueh Chang, Hsiu-Chuan Wu, Ting-Yu Chang, Chien-Yu Lin, Yeu-Jhy Chang.

**Project administration:** Yeu-Jhy Chang.

**Resources:** Kang-Hsing Fan.

**Supervision:** Chien-Hung Chang.

**Writing – original draft:** Chi-Hung Liu, Joseph Tung-Chieh Chang.

**Writing – review & editing:** Chi-Hung Liu, Joseph Tung-Chieh Chang, Tsong-Hai Lee, Yeu-Jhy Chang.

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
