## [Decision Letter · Decision Letter 0]

9 Dec 2020

PONE-D-20-34151

Total Plaque Score Helps Determining Follow-up Strategy for Carotid Artery Stenosis Progression in Head and Neck Cancer Patients after Radiation therapy

PLOS ONE

Dear Dr. Chang,

Thank you for submitting your manuscript to PLOS ONE. After careful consideration, we feel that it has merit but does not fully meet PLOS ONE’s publication criteria as it currently stands. Therefore, we invite you to submit a revised version of the manuscript that addresses the points raised during the review process.

The manuscript will need extensive English editing.

We look forward to receiving your revised manuscript.

Kind regards,

Paula Boaventura, PhD

Academic Editor

PLOS ONE

Additional Editor Comments:

The manuscript will need extensive English editing.

Journal Requirements:

Reviewers' comments:

Reviewer's Responses to Questions

**Comments to the Author**

1. Is the manuscript technically sound, and do the data support the conclusions?

Reviewer #1: Yes

Reviewer #2: Partly

Reviewer #3: Partly

2. Has the statistical analysis been performed appropriately and rigorously? 

Reviewer #1: Yes

Reviewer #2: Yes

Reviewer #3: Yes

3. Have the authors made all data underlying the findings in their manuscript fully available?

Reviewer #1: No

Reviewer #2: Yes

Reviewer #3: Yes

4. Is the manuscript presented in an intelligible fashion and written in standard English?

Reviewer #1: Yes

Reviewer #2: Yes

Reviewer #3: Yes

5. Review Comments to the Author

Reviewer #1: The authors sought to investigate if carotid duplex ultrasound (CDU) predicts the development of ischemic stroke (IS) in patients with mild carotid artery stenosis after radiation therapy (RT) for head and neck cancer (HNC). They prospectively studied all patients treated with RT at their hospital so the population is equivalent to a random sample. They utilized a total plaque score (TPS) to categorize participants as low plaque (LP) and high plaque (HP). Utilizing multivariate logistic regression the authors determined that TPS was the main predictor of carotid arterial disease (CAD) progression. They noted that in contrast to the LP group, membership in the HP group was associated with older age, male sex, hypertension, smoking, impaired renal functioning, antiplatelet use, and increased uric acid The authors conclude CDU may be a valuable tool to assess patients with HNC immediately after RT and to follow those with HP regularly to consider for intervention if there is progression of CAD. The study was well-conducted with a reasonable analysis. The manuscript will likely merit publication with revisions.

Page 4

The authors state that all data are fully available without restriction. Please publish the full dataset without identifying items to accompany the manuscript. If the datasets are too large to be supplementary files, then please publish in a separate repository, such as Zenodo or Mendeley.

Page 7

Keywords

Since some search engine list only the title and keywords, choose terms that are not included in the title.

Page 17

Second paragraph

First line

The authors note that HNC is common in Chinese populations. How many of their participants had Han ancestry? Does Han ancestry play a role in the outcomes if introduced to their models? Please identify the Han Chinese heritage of each participant to include in the multivariate logistic regression models.

Figure 3.

Legend

Define "cum" ordinate.

Reviewer #2: thank you to the authors for their submission. This is a cogent, nicely composed, well rationalized, and potentially informative study intending to assess the predictive value of readily available ultrasonographic evaluation of patients with radiation induced carotid injury following HnN cancer. I would applaud the authors for attempting to study a widely available, easily repeatable, and, generally speaking, affordable methodology for this purpose. I do see very strong potential for its applications, but a few criticisms come to mind which I do believe are likely addressable. In all I believe the study would be of interest of the readership.

I would emphasize the following:

1. I do believe some ground truth comparison to cross-sectional angiographic methods that offer a more legitimate structural assessment would have been a complement. I do understand that this may not be possible in the entirety of the cohort, but even if a subset has such imaging, whether CTA,

cath, or MRI , it would greatly contribute

2. I believe the longitudinal follow up is still Somewhat short term, and truly disambiguating in the arc of disease in different patients would require more regular and longer-term evaluation. I understand that this may be considered hypothesis building data for know and I hope the authors can emphasize that. Completely shifting practice paradigms based on what our somewhat preliminary results should be done only with immense circumspection

Lastly, imaging examples are completely lacking. Keep in mind that this is a highly operator dependent technique, and if anything I would suggest a large number of imaging examples including those that might highlight pitfalls or limitations.

Reviewer #3: The authors present methodology to predict progression of carotid arterial stenosis in head and neck cancer patients who have undergone radiation therapy.

The authors report the results of 217 such patients who have undergone carotid doppler ultrasound. The studies were reviewed by blinded trained radiologists. The inter-rater reliability was > 90% agreement. (HOW WERE THE REMAINING DISAGREEMENTS HANDLED).

The authors report that high initial plaque score of >/= to 7 is very important in determining likelihood of progression in carotid stenosis following radiation therapy. (PLEASE REPORT THE TRANSDUCER ANGLE AT WHICH THE VELOCITIES WERE ACQUIRED FOR THE INTIIAL WORK UP AND FOR THE FOLLOW-UP SESSION. DIFFERENCES IN THE ANGLE COULD ARTIFICIALLY ELEVATE THE MEASURED ENDOVASCULAR FLOW VELOCITIES AND ERRONEOUSLY LEAD ONE TO INTERPRET INCREASING STENOSIS. PLEASE PROVIDE THIS INFORMATION.

6. PLOS authors have the option to publish the peer review history of their article (what does this mean?). If published, this will include your full peer review and any attached files.

Reviewer #1: **Yes: **James Robert Brasic

Reviewer #2: No

Reviewer #3: No

---

## [Author Response · Author response to Decision Letter 0]

25 Dec 2020

Dear Paula Boaventura

Thank you for your nice and detailed review. We’ve provided a point-by-point revision and our responses to all of your comments. The reasons and revisions are provided in the following contents. In the revised manuscript, all the changes are highlighted. We deeply appreciate your valuable review, which stimulated a more thorough consideration of the essay. Thank you very much.

Sincerely Yours,

Yeu-Jhy Chang, MD

Stroke Center and Department of Neurology

Chang Gung Memorial Hospital, Linkou Medical Center and College of Medicine, Chang Gung University, Taoyuan, Taiwan

No. 5, Fu-Hsing ST. Kueishan, Taoyuan, 33333 Taiwan

Tel: 886-3-3281200 ext 8340 

Fax: 886-3-3288849

Editor’s comment

1. The manuscript will need extensive English editing.

Answer: Thank you, we’ve tried our best to improve the English editing, including the English revision by the native speaker.

Answer: Thank you, we’ve revised the manuscript according to PLOS ONE's style requirements.

Answer: Thank you for your reminding. We’ve added this in the manuscript.

Materials and Methods, page 4, line 12-14

The study was approved by the Ethics Institutional Review Board of Chang Gung Memorial Hospital (IRB No. 100-4153B). All of the included patients signed written informed consent forms.

Reviewers' comments:

Reviewer #1: The study was well-conducted with a reasonable analysis. The manuscript will likely merit publication with revisions.

1. The authors state that all data are fully available without restriction. Please publish the full dataset without identifying items to accompany the manuscript. If the datasets are too large to be supplementary files, then please publish in a separate repository, such as Zenodo or Mendeley.

Answer: Thank you for your comment. After discussing with the Ethics Institutional Review Board of out hospital, we’ve provided the data those could be accessible in the supplementary file.

2. Keywords

Since some search engine list only the title and keywords, choose terms that are not included in the title.

Answer: Thank you for your wonderful suggestion. We’ve added one keyword.

Keywords: Carotid artery stenosis, carotid duplex ultrasound, head and neck cancer, plaque scores, radiation therapy, radiation vasculopathy.

3. The authors note that HNC is common in Chinese populations. How many of their participants had Han ancestry? Does Han ancestry play a role in the outcomes if introduced to their models? Please identify the Han Chinese heritage of each participant to include in the multivariate logistic regression models.

Answer: Thank you for your expert opinion. In Taiwan, more than 95 percent of the population claiming Han ancestry (https://www.taiwan.gov.tw/content_2.php). In our study, all recruited patients were Han Chinese. It would be difficult to discuss the role of Han Chinese heritage in clinical outcomes.

We revised the manuscript accordingly.

Results, page 7, line 23-24

 Between 2013 and 2014, we prospectively screened 361 Han Chinese patients with HNC who received RT at our hospital.

Discussion, page 12, line 12-14

We stated this in the limitation paragraph.

Fourth, the single-center nature of this study could have led to patient selection bias. Fifth, the generalizability of our conclusions to other ethnicities remains uncertain.

4. Figure 3. Legend Define "cum" ordinate.

Answer: Thank you for your suggestion. We’ve revised the ordinate of this figure. Besides, due to one figure was suggested to be added by another reviewer, figure 3 was renamed as figure 4.

New ordinate of this figure is “Cumulative CAS progression-free rate”.

Figure 4. Multivariate-adjusted survival curves estimating the time to CAS progression. The incidence rate of CAS progression was significantly higher in the HP group than in the LP group. 

CAS, carotid artery stenosis; HP, high plaque score; LP, low plaque score; TPS, total plaque score.

Reviewer #2: This is a cogent, nicely composed, well rationalized, and potentially informative study intending to assess the predictive value of readily available ultrasonographic evaluation of patients with radiation induced carotid injury following HnN cancer. I would applaud the authors for attempting to study a widely available, easily repeatable, and, generally speaking, affordable methodology for this purpose. I do see very strong potential for its applications, but a few criticisms come to mind which I do believe are likely addressable. In all I believe the study would be of interest of the readership.

Answer: Thank you for your suggestion. We will keep working hard.

1. I do believe some ground truth comparison to cross-sectional angiographic methods that offer a more legitimate structural assessment would have been a complement. I do understand that this may not be possible in the entirety of the cohort, but even if a subset has such imaging, whether CTA, cath, or MRI , it would greatly contribute

Answer: Thank you for your suggestion. These are indeed important issues. However, due to the limitation of this study, we could only address this in the limitation paragraph. In the future study, we will endorse this suggestion into study protocol.

Discussion, page 11, line 26- page 12, line 9

First, CDU studies may not be a precise method to evaluate the degree of stenosis and plaque features. Confounding factors caused by the technique may have influenced the study results, and thus limited the interpretation of CAS progression. In addition, advanced cerebrovascular images including computerized tomography angiography, magnetic resonance angiography, or conventional angiography were not routinely arranged as a confirmatory study when CAS progression was detected by CDU. This may also have limited the interpretation of the study results. Second, baseline vascular status may have confounded the conclusions drawn from our results. A higher TPS or CAS before RT may have further worsened the clinical progression and may have led to bias in this study. However, in real-world practice, primary physicians may not know the vascular status of the carotid arteries prior to RT. Further cohort studies with more strict follow-up protocols and confirmatory vascular imaging studies are still needed to validate our suggested follow-up strategy. 

2. I believe the longitudinal follow up is still Somewhat short term, and truly disambiguating in the arc of disease in different patients would require more regular and longer-term evaluation. I understand that this may be considered hypothesis building data for know and I hope the authors can emphasize that. Completely shifting practice paradigms based on what our somewhat preliminary results should be done only with immense circumspection

Answer: Thank you for your suggestion. These are quite important. We’ve added this into the limitation paragraph.

Discussion, page 12, line 14-16

Lastly, a longer follow-up duration may provide more conclusive answers. Completely shifting practice paradigms based on what our preliminary results should be done only with immense circumspection.

3.Lastly, imaging examples are completely lacking. Keep in mind that this is a highly operator dependent technique, and if anything I would suggest a large number of imaging examples including those that might highlight pitfalls or limitations.

Answer: Thank you for your suggestion. These are quite important. We’ve added one figure (figure 2) showing imaging examples.

Results, page 8, line 3-5

Of these 217 patients, 209 (96.3%) received CDU follow-up for more than 18 months after enrolment. We first recorded the presence and evaluated the predictors of CAS progression (Figure 2).

Figure 2. An illustrated case with carotid artery stenosis progression.

This figure shows typical carotid duplex ultrasound findings in a patient after radiation therapy. Prominent segmental changes were noted in bilateral common carotid arteries at the time of enrolment (A-C). Follow-up carotid duplex ultrasonography of the same patient 2 years later (D-F) demonstrated significant progression of stenosis (>50%) in the left common carotid artery (D) with hemodynamic changes (E). 

Reviewer #3: The authors report the results of 217 such patients who have undergone carotid doppler ultrasound. The studies were reviewed by blinded trained radiologists. The inter-rater reliability was > 90% agreement. 

1. (HOW WERE THE REMAINING DISAGREEMENTS HANDLED).

Answer: thank you for your comment.

The results of the CDU were read by single blind neurologist (CLC) when performing this retrospective analysis. Dr. CLC did not know the status of each enrolled patient. We compared Dr. CLC’s CDU results to the official reports of the CDU documented at the time of examination (between January 1, 2013 and December 31, 2014). There was no discordance between the official reports and Dr. CLC’s documentation. We stated this in the revised manuscript. Besides, “> 90%” was not the “inter-rater reliability” of CDU reporting. Instead, “> 90%” was the accuracy data comparing CDU results and conventional angiography when we set-up our CDU laboratory in 1992. We added the reference (Ref. 16) for this accuracy data.

Materials and Methods, page 5, line 19-23

The investigators who performed and read (CLC et al.) the baseline and follow-up CDU were blinded to the patients’ clinical conditions. Agreement was achieved in all CDU results between the investigators. Our CDU laboratory has an overall diagnostic accuracy rate of >90% for carotid stenosis [15, 16].

Newly added reference: 

16. Tseng KY, Lee TH, Ryu SJ, Chen ST. Correlation between sonographic and angiographic findings of extracranial carotid artery disease. Zhonghua yi xue za zhi = Chinese medical journal; Free China ed. 1992;50(4):302-6. Epub 1992/10/01. PubMed PMID: 1334789.

2. The authors report that high initial plaque score of >/= to 7 is very important in determining likelihood of progression in carotid stenosis following radiation therapy. (PLEASE REPORT THE TRANSDUCER ANGLE AT WHICH THE VELOCITIES WERE ACQUIRED FOR THE INTIIAL WORK UP AND FOR THE FOLLOW-UP SESSION. DIFFERENCES IN THE ANGLE COULD ARTIFICIALLY ELEVATE THE MEASURED ENDOVASCULAR FLOW VELOCITIES AND ERRONEOUSLY LEAD ONE TO INTERPRET INCREASING STENOSIS. PLEASE PROVIDE THIS INFORMATION.

Answer: Thank you for your reminding, we’ve revised the manuscript accordingly.

Materials and Methods, page 5, line 17-18

The degree of CAS was examined in the CDU study according to standard ultrasound criteria. The percentage of maximum stenosis in longitudinal views was determined using computer-assisted measurements of the 1-residual lumen diameter/vessel diameter × 100. The angle of insonation was set at 60 degrees during the flow velocity examination. Peak systolic velocities ≥120 cm/s were used to identify ≥50% CAS [14].

Thank you for the reviewers’ constructive and valuable advice and comments.

We hope the revised version could be more readable and can meet the requirements of PLOS ONE.

Best regards

Yeu-Jhy Chang, MD

Stroke Center and Department of Neurology

Chang Gung Memorial Hospital, Linkou Medical Center and College of Medicine, Chang Gung University, Taoyuan, Taiwan

---

## [Decision Letter · Decision Letter 1]

20 Jan 2021

PONE-D-20-34151R1

Total Plaque Score Helps to Determine Follow-up Strategy for Carotid Artery Stenosis Progression in Head and Neck Cancer Patients after Radiation Therapy

PLOS ONE

Dear Dr. Chang,

Thank you for submitting your manuscript to PLOS ONE. After careful consideration, we feel that it has merit but does not fully meet PLOS ONE’s publication criteria as it currently stands. Therefore, we invite you to submit a revised version of the manuscript that addresses the points raised during the review process.

We look forward to receiving your revised manuscript.

Kind regards,

Paula Boaventura, PhD

Academic Editor

PLOS ONE

Reviewers' comments:

Reviewer's Responses to Questions

**Comments to the Author**

1. If the authors have adequately addressed your comments raised in a previous round of review and you feel that this manuscript is now acceptable for publication, you may indicate that here to bypass the “Comments to the Author” section, enter your conflict of interest statement in the “Confidential to Editor” section, and submit your "Accept" recommendation.

Reviewer #1: (No Response)

Reviewer #2: All comments have been addressed

Reviewer #3: All comments have been addressed

2. Is the manuscript technically sound, and do the data support the conclusions?

Reviewer #1: Yes

Reviewer #2: Yes

Reviewer #3: Yes

3. Has the statistical analysis been performed appropriately and rigorously? 

Reviewer #1: Yes

Reviewer #2: Yes

Reviewer #3: Yes

4. Have the authors made all data underlying the findings in their manuscript fully available?

Reviewer #1: Yes

Reviewer #2: Yes

Reviewer #3: Yes

5. Is the manuscript presented in an intelligible fashion and written in standard English?

Reviewer #1: Yes

Reviewer #2: Yes

Reviewer #3: Yes

6. Review Comments to the Author

Reviewer #1: The authors assessed the utility of a total plaque score to predict carotid artery stenosis progression in patients with head and neck cancer after radiation.

The study was well-conducted with a reasonable analysis.

The manuscript will likely merit publication with revision to facilitate interpretation by clinicians.

Page 28

Figure 2

Legend

Please state the age and sex of the participant.

Reviewer #2: Thank you for making the requested additions and clarifications. and its current form of this manuscript, while in park comprising preliminary or hypothesis building data is likely to offer some valuable insights regarding the intersection of credit disease and radiation – treated head and neck cancer patients. The largest areas for future clarification are cross reference with advanced structural cross-sectional angiographic imaging, the presence of longitudinal data to assess true progression over a longer period of time, and a better understanding of baseline variables in these patients.

Reviewer #3: My concerns about potentially suboptimal transducer angles contributing to elevated endovascular velocities was satisfactorily addressed.

7. PLOS authors have the option to publish the peer review history of their article (what does this mean?). If published, this will include your full peer review and any attached files.

Reviewer #1: **Yes: **James Robert Brasic

Reviewer #2: No

Reviewer #3: No

---

## [Author Response · Author response to Decision Letter 1]

20 Jan 2021

Reviewers' comments:

Reviewer #1: The authors assessed the utility of a total plaque score to predict carotid artery stenosis progression in patients with head and neck cancer after radiation.

The study was well-conducted with a reasonable analysis.

The manuscript will likely merit publication with revision to facilitate interpretation by clinicians.

Page 28 Figure 2 Legend

Please state the age and sex of the participant.

Answer: Thank you for your suggestion. We’ve added the age and sex of the illustrated case.

Figure legends, figure 2

This figure shows typical carotid duplex ultrasound findings in a 69-year-old male patient after radiation therapy.

Reviewer #2: Thank you for making the requested additions and clarifications. and its current form of this manuscript, while in park comprising preliminary or hypothesis building data is likely to offer some valuable insights regarding the intersection of credit disease and radiation – treated head and neck cancer patients. The largest areas for future clarification are cross reference with advanced structural cross-sectional angiographic imaging, the presence of longitudinal data to assess true progression over a longer period of time, and a better understanding of baseline variables in these patients.

Answer: Thank you for your kind comment, which indeed help us to design a better study to answer this question. We’ve revised the last paragraph of our discussion accordingly.

Discussion, page 12, line 17-20

Completely shifting practice paradigms based on what our preliminary results should be done only with immense circumspection. In the future, we may need a cross reference with advanced structural cross-sectional angiographic imaging, a longitudinal data to assess true progression over a longer period of time, and more comprehensive baseline variables in these patients for further clarification of our conclusions. 

Reviewer #3: My concerns about potentially suboptimal transducer angles contributing to elevated endovascular velocities was satisfactorily addressed.

Answer: Thank you for your wonderful help. Your excellent comments indeed make our manuscript more readable.

---

## [Editor Report · Decision Letter 2]

25 Jan 2021

Total Plaque Score Helps to Determine Follow-up Strategy for Carotid Artery Stenosis Progression in Head and Neck Cancer Patients after Radiation Therapy

PONE-D-20-34151R2

Dear Dr. Chang,

We’re pleased to inform you that your manuscript has been judged scientifically suitable for publication and will be formally accepted for publication once it meets all outstanding technical requirements.

Kind regards,

Paula Boaventura, PhD

Academic Editor

PLOS ONE
---

## [Editor Report · Acceptance letter]

2 Feb 2021

PONE-D-20-34151R2 

Total Plaque Score Helps to Determine Follow-up Strategy for Carotid Artery Stenosis Progression in Head and Neck Cancer Patients after Radiation Therapy 

Dear Dr. Chang:

I'm pleased to inform you that your manuscript has been deemed suitable for publication in PLOS ONE. Congratulations! Your manuscript is now with our production department. 

Kind regards, 

on behalf of

Dr. Paula Boaventura 

Academic Editor

PLOS ONE